# Fast and Robust Visual Tracking with Few-Iteration Meta-Learning

**DOI:** 10.3390/s22155826

**Published:** 2022-08-04

**Authors:** Zhenxin Li, Xuande Zhang, Long Xu, Weiqiang Zhang

**Affiliations:** 1School of Electronic Information and Artificial Intelligence, Shaanxi University of Science and Technology, Xi’an 710021, China; 2State Key Laboratory of Space Weather, National Space Science Center, Chinese Academy of Sciences, Beijing 100190, China; 3Peng Cheng National Laboratory, Shenzhen 518000, China; 4School of Mathematics and Statistics, Shenzhen University, Shenzhen 518060, China

**Keywords:** object tracking, meta-learning, few iterations, robustness, real-time, transformer

## Abstract

Visual object tracking has been a major research topic in the field of computer vision for many years. Object tracking aims to identify and localize objects of interest in subsequent frames, given the bounding box of the first frame. In addition, the object-tracking algorithms are also required to have robustness and real-time performance. These requirements create some unique challenges, which can easily become overfitting if given a very small training dataset of objects during offline training. On the other hand, if there are too many iterations in the model-optimization process during offline training or in the model-update process during online tracking, it will cause the problem of poor real-time performance. We address these problems by introducing a meta-learning method based on fast optimization. Our proposed tracking architecture mainly contains two parts, one is the base learner and the other is the meta learner. The base learner is primarily a target and background classifier, in addition, there is an object bounding box prediction regression network. The primary goal of a meta learner based on the transformer is to learn the representations used by the classifier. The accuracy of our proposed algorithm on OTB2015 and LaSOT is 0.930 and 0.688, respectively. Moreover, it performs well on VOT2018 and GOT-10k datasets. Combined with the comparative experiments on real-time performance, our algorithm is fast and robust.

## 1. Introduction

As the main component of the computer vision system, visual object tracking (VOT) is also a hot issue in current research. The purpose of VOT is to obtain the initial target-state information in the first frame for a given video image sequence, which is mainly the target bounding box obtained by manual calibration or object detection. Pattern recognition, pre-trained deep network models, digital image processing, and other related technologies can be used to estimate the target-state information in subsequent frames, including the position of the target center point of the target and the shape of the target, that is, the size of the scale. In recent years, with the improvement of the quality of camera terminals, the speed of computer processing of data, and the increasing demand for image processing, visual object tracking algorithms have attracted more and more researchers’ attention. It is widely used in many practical scenarios, such as video surveillance [1,2], intelligent transportation [3], medical diagnosis [4], and military applications [5].

Robustness and real-time performance are two important factors for VOT, and the definitions of both are described below [6]. We describe robustness in target tracking as the ability of a tracking algorithm to be accurate under environmental conditions that would degrade performance; for instance, partial occlusion, photometric changes, incorrect edge matching, and so on. In addition, tracking speed is another very important metric for evaluating trackers, especially to meet real-time requirements. However, there are many key factors influencing tracking speed, including feature extraction, model update methods, programming language, and hardware implementation of the tracker. According to the above description, designing a robust tracker, i.e., a tracker which is highly accurate and real-time, is still a challenging task.

Recently, deep learning and convolutional neural networks are increasingly used in VOT due to their rich representation and generalization capabilities. At the same time, Siamese-based trackers have gained attention in recent years because of their balanced speed and accuracy.

In this paper, we choose a recent single-object tracking method based on the Siamese network and transformer as the baseline for this work [7]. Since the existence of rich temporal contextual information between consecutive frames is neglected in the original tracker for visual object tracking tasks, this baseline work introduces transformer in Siamese network and discriminative correlation filter networks, respectively, to improve the powerful tracking potential of the simple algorithmic model by fully exploiting temporal information. It has been experimentally shown that the baseline algorithm has achieved state-of-the-art (SOTA) performance. 

However, the above algorithm still suffers from some common problems of VOT. First, in the tracking process, if there is severe occlusion, the predicted target bounding box cannot accurately locate the target; second, in the initialization and optimization of the classification regression model, the training time is too long due to too many iterations and high computation; third, in the online tracking, there is no reasonable online update mechanism, which leads to the need for tracking robustness still to be improved.

In this paper, to address this issue, a flexible few-iteration meta-learning framework based on an efficient unrolled optimization strategy is introduced to speed up the process of initializing a categorical regression model [8]. Meta-learning itself involves different approaches. Two methods have proven to be very effective: metric learning and optimization-based approach [9]. The first aims to learn the embedding space, and the second aims to optimize the model parameter set to solve a specific task. Our proposed tracking network framework is built on an optimization-based paradigm, we have mainly considered the design of two modules: a meta learner and a base learner. The meta learner generally learns a feature representation across template branch and search branch. The base learner, on the other hand, performs the task-specific adaptation—classification of target and background and regression of object bounding box.

In summary, the four contributions of this article are listed in the following:(1)We proposed a novel classification and regression model initialization procedure during offline training of single object tracking networks, which is based on an optimization meta-learning approach;(2)We considered that the features required for the classification and regression tasks of single visual object tracking are not identical, so two branches are used to obtain the final results;(3)We innovatively applied a new optimization-based meta-learning method in the field of single visual object tracking;(4)We proposed a novel template online update mechanism to the Siamese object tracking network to improve the accuracy of a tracker.

To validate the effectiveness of our proposed framework, we compare our method to other state-of-the-art trackers on several popular benchmark datasets, including OTB2015 [10], VOT2018 [11], LaSOT [12], and GOT-10k [13].

The whole article consists of five chapters. The first chapter, as an introduction, introduces the research background and motivation of this work. The second chapter is related work, which introduces the classification of visual object tracking, object tracking based on meta-learning, and the update mechanism in tracking network. The third chapter describes the network sub-module proposed in this paper in detail. The fourth chapter is the details and results analysis of the experiment. The fifth chapter is the summary and outlook.

## 2. Related Work

### 2.1. Deep Neural Network-Based Tracking Algorithms

In the last decade, Discriminative Correlation Filters (DCF) and Siamese Networks (SN) have become the two most dominant network architectures in the field of tracking [14]. In the following, the tracking algorithm of both frameworks is briefly described.

DCF is a supervised technique for learning a linear regressor. DCF learns a correlation filter online to localize the target object in subsequent frames by minimizing a Least Squares Output Error. The earliest MOSSE [15] algorithm utilized only single-channel grayscale features, later, CSK [16] and KCF [17] fused multi-channel features, then, people considered color features and proposed STAPLE [18], and, finally, algorithms such as DeepSRDCF [19], HCF [20], ECO [21], and UPDT [22] started to utilize depth features. In addition to this, some of the top performing trackers in recent years have actually been in this area, such as ATOM [23], DiMP [24], and PrDiMP [25].

SN-based tracker is two parallel networks with the same or similar structure, which transforms the target tracking problem into a matching problem for a given template and candidate images. SiamFC [26], GOTURN [27], and SINT [28], are three representatives of the earliest pioneering work using Siamese networks. To achieve more accurate localization and scale estimation, SiamRPN [29] introduced a region proposal network in object detection. Immediately afterward, DaSiamRPN [30] enhanced the discriminative ability of the tracker by introducing negative pairs to suppress the interference factors during training. Unlike the former, the anchor-free regression has also received attention and thus applied to the field of the VOT, with representative algorithms, such as SiamBAN [31], Ocean [32], and SiamCAR [33]. The latest research, i.e., TransT [34] and TrDiMP [7], also incorporates the hottest transformer structure into the field.

The SN-based trackers and DCF trackers have their advantages, but the tracker based on the Siamese network has attracted our attention because it can balance accuracy and speed. However, the most serious challenges are that SN-based trackers also have some limitations in terms of a large number of labeled images required for offline training, feature extraction networks, the lack of an effective online update mechanism, design of loss function, and influence of scale change on tracking accuracy. All these factors affect the performance of the tracker, so, finding a better way to make the tracker more robust and fast becomes another challenge.

### 2.2. Meta-Learning for Visual Tracking 

The ability to learn from one or several instances is a fundamental characteristic of human intelligence; for example, a person who has seen a dog for the first time can quickly identify it from different animals even afterward, whereas a machine needs a lot of data for training to distinguish between different classes of animals. This property is known as meta-learning, or learning to learn [35]. It can be broadly classified into two main categories, metric-based and optimization-based.

With the success of meta-learning in classification tasks, more and more researchers have started to introduce meta-learning to the field of VOT. Meta-Tracker [36] was the first to apply meta-learning to some trackers. The tracking results of MDNet [37] and CREST [38] are improved by training a more robust feature initialization extractor using the meta-learning approach of MAML [39]. Wang et al. [40] view tracking as a special detection problem and use meta-learning to train the detectors FCOS and RetinaNet on the tracking sequences, allowing the detectors to localize the target better with only a few steps of gradient descent from a single frame, thus greatly improving the real-time performance of tracking. It can be seen that a reasonable introduction of meta-learning methods into the target tracking algorithm can effectively improve the robustness and real-time performance of the algorithm.

These methods mentioned above are based on the idea of meta-optimization, so we also try to apply such methods to visual object tracking based on the Siamese network.

### 2.3. Online Model Update

In many SN-based single object tracking algorithms, the target template is initialized in the first frame and then remains fixed for subsequent frames of the video, and the tracker does not perform any template updates, so the tracking performance is completely dependent on the general matching ability of the SN.

Article [41] uses the current context to learn a new model as an adaptation to change, multiple features provided by spatio-temporal information are used to enhance the target representation. An end-to-end tracking architecture [24] is used that makes full use of information from the target and the background for target model prediction, by designing an iterative optimization process to obtain discriminative learning loss. Using the information from the gradient for template updating [42], the two subnetworks in GradNet use the discriminative information from the gradient by forward and reverse operations. A visual tracking algorithm [43] based on a deep meta-learning network is constructed in the target feature space, and the algorithm generates target-adaptive weights through the meta-learning network to adapt to the target appearance.

While many techniques have been proposed for template updates, simply not using any updates is still surprisingly robust. However, to meet the real-time requirement, we were inspired by [24,43] to design a new online update method that achieves higher FPS without reducing the accuracy.

## 3. Proposed Method

In this work, we propose a visual object tracking network based on the Siamese network and meta-learning. As shown in Figure 1, the target tracking network architecture proposed in this paper consists of the following main parts: a part called meta-learner, which consists of a backbone network used for feature extraction, and an encoder in the transformer used for feature encoding. The other part is called base-learner, which is composed of a discriminative classification sub-network and a regression sub-network with a target bounding box.

In Section 3.1, we first review the transformer-assisted tracking framework and analyze the limitations of the method. Next, we describe the meta-learner in Section 3.2, Section 3.3 discusses the base learner, i.e., target classification and bounding box regression, in detail. Finally, the offline training process and online visual tracking are described in Section 3.4 and Section 3.5, respectively.

### 3.1. Background

One of the popular paradigms for VOT is deep Siamese network-based tracking. The paradigm is to train the model end-to-end offline using a large amount of labeled data and treat it as a similarity learning problem. Figure 2 illustrates the architecture of a simple SN network.

As shown in Figure 2, the SN network includes a template branch and a search branch with the same backbone parameters in both branches. After a pair of pictures go through two branches to get a feature map, they are matched using a cross-correlation operation, that is, where the two branches meet in the graph. The main purpose of SN is to improve the limitations of the original deep CNN using pre-training, which makes full use of end-to-end learning for real-time tracking. Offline training is meant to be used to guide the tracker through complex challenges, such as rotation, lighting changes, etc. The tracker uses SN to learn the relationship between object motion and appearance and, thus, is used to locate targets that have not been seen in training.

After continuous research and improvement of numerous algorithms, Wang et al. proposed a new architecture called TrSiam and TrDiMP [7]. The authors introduced a carefully modified transformer into the SN-based tracking framework and the DCF framework, which connects interval frames in consecutive video frames and passes rich temporal information between frames that can effectively improve tracking accuracy.

To design a more robust and faster tracking framework and to apply the Few-Iteration Meta-Learning method to the whole network, TrSiam becomes a candidate for our algorithm to be improved.

### 3.2. Meta-Learner with Feature Extractor

As shown in the dotted box in Figure 1, the first part of our model is the meta-learner, which follows the basic architecture of Siamese networks and is composed of two parallel branches of the backbone with shared weights, which are the template branch and the search branch. Following the architectural design analysis in [44], we use ResNet-50 as the backbone network. Next, we split the encoder and decoder of the transformer into two branches to accommodate the Siamese tracking method; the structure of the encoder and decoder is shown in Figure 3.

The meta-learner M trains a generic feature model with the aim of performing well on unseen tracking sequences. The goal of meta-learner M is to learn a rich feature representation, and then the base learner Β can classify targets and backgrounds of the tracked video frames on the feature representation. The meta-learning process can be represented by the following equation:(1)$$M\left(D\right):=\phi^{*}=\underset{\phi }{\mathrm{argmin}\;}E_{M_{0}\sim M}L_{meta}\left(M_{0},m_{\phi };b_{\theta^{*}}\right)$$
where *M* is a series of classification problems that classify the target from the search region that sampled from training datasets D, $M_{0}$ denotes a video sequence where the first few frames are used as input to the template branch and the subsequent frames are used as input to the search branch, and $L_{meta}$ represents the meta-training loss, which choose the cross-entropy loss. $m_{\phi }$ and $b_{\theta }$ represent meta-learner and base-learner, respectively.

Our meta-learner inputs are pairs of sets $\left(M_{template},\;M_{search}\right)$. Each set $M={\left\{\left(I_{i},b_{i}\right)\right\}}_{i=1}^{N}$ consists of an image $I_{i}$ and the corresponding target bounding box $b_{i}$ with it.

In template branch, several frames (i.e., the template image patch z∈ℝ3×H0×W0) are first passed to the backbone network to obtain their features maps fz∈ℝC×H×W, and then fed template feature map to the encoder for feature representation. The basic block of the encoder is the self-attention mechanism [45], which aims to mutually enforce the features from multiple templates. As shown in Figure 3, there are three inputs to this basic block, namely value V, query Q, and key K.

Given a set of template features maps fzi with a spatial size of H×W and dimensionality *C*, the self-attention block is used to calculate the similarity matrix AT→T. Let Ti=fzi, T=ConcatT1,…, Tn∈ℝn×C×H×W, the AT→T matrix can be computed as equation:(2)AT→T=AttenT,T=AttenφT′,φT′=SoftmaxcolφT′¯ φT′¯T/τ
where T′ is reshaped from *T* (i.e., T′∈ℝNT×C,NT=n×H×W), φ· is a 1×1 linear transformation that reduces the embedding channel from *C* to *C/4*. Where ·¯ is ℓ2—normalized feature across the channel dimension, and τ is a temperature parameter controlling the Softmax distribution.

**Figure 3 sensors-22-05826-f003:**
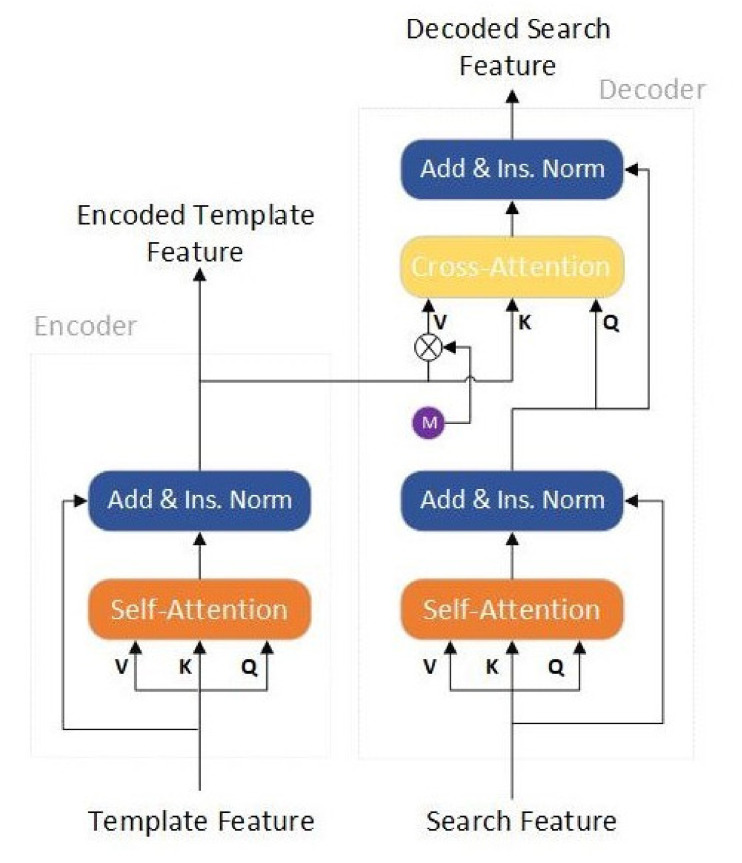
Simplified transformer architecture for Siamese-based tracking networks.

After a self-attention module, a residual block, and an instance normalization module immediately follow. The encoded template feature T^∈ℝNT×C can be calculated by this part.
(3)T^=Ins.NormAT→TT′+T′
where Ins.Norm(·) denotes the instance normalization.

In the encoder structure, the three values of *V*, *K*, and *Q* correspond to *T*′, *T*, and *T*, respectively. After our simplified encoder, a high-quality feature representation T^ is obtained from several template images, which is also need to fed to the decoder to reinforce the search patch feature. Besides, T^ is also reshaped back to Tz∈ℝn×C×H×W, which acts the template CNN kernel to convolve with Sx for response generation.

Another important point is that we also need to focus on the target position in the template. As with baseline, we use the Gaussian function to generate a mask of template features.
(4)$$m\left(y\right)=exp\left(-\frac{{\left|\left|y-c\right|\right|}^{2}}{2\sigma^{2}}\right)$$
where *c* is the ground truth target position. Also, in order to maintain the consistency of the calculation,
(5)M=Concatm1,…, mn∈ℝn×H×W

In search branch, the search region image patch x∈ℝ3×H0×W0, similar to the template branching process. First, the search feature map fx∈ℝC×H×W goes to the self-attentive module of the decoder to calculate the similarity matrix Afx→fx. For ease of representation, we let S=fx, S′∈ℝNS×C, and NS=H×W. Then, we transform the search feature through AS→SS′. As a result, S0^ is obtained again by residual concatenation and normalization.
(6)S0^=Ins.NormAs→sS′+S′

Similarly, in the decoder structure, the three values of *V*, *K*, and *Q* correspond to *S**′*, *S*, and *S*, respectively.

In the next step, we use the encoded template feature $\hat{T}$ and mask M obtained from the encoder, and $\hat{S_{0}}$ as input to the Cross-Attention module. This is done for two reasons: first, by introducing mask, potential target regions can be better highlighted; the other is to integrate the contextual information of template features into the search features. Therefore, the final search-decoding features can be calculated using the following equation:(7)$$\hat{S}=\mathrm{Ins}.\mathrm{Norm}\left(A_{T\to S}\left(\hat{T}⊗M^{\prime }\right)+\hat{S_{0}}\right)$$
where ⊗ is the element-wise multiplication and $M^{\prime }$ is obtained by flattening M. By using transformer’s spatial attention mechanism, the search feature $\hat{S_{0}}$ better highlights the potential object search area. One difference is that at this time, the input *V*, *K*, and *Q* represent $\hat{T}⊗M^{\prime }$, *T*, and *S*, respectively.

### 3.3. Base-Learner with Classification and Regression

In this subsection, we describe the base learner Β. We consider the tracking task as a classification task with target and background and a regression task with target bounding box. Afterwards, the inference under this interpretation is analyzed in detail and simplified to obtain speed-ups.

Trained on a given dataset, our base learner aims to learn a classifier $b_{\theta }$, the representation learned by the meta learner is used as input, which is a simplified version of our proposed transformer. For a search image patch x, the classifier in the base network will output a predicted value of the classification score with the template branch. The general form of the base learner Β can be given by this formula.
(8)$$Β\left(M_{0};m_{\phi }\right):=\theta^{*}=\underset{\theta }{\mathrm{argmin}}\;L_{\mathrm{base}}\left(M_{0},b_{\theta };m_{\phi }\right)$$

The base learner Β minimizes $L_{\mathrm{base}}$ with regard to the parameters θ. The output of the classifier, where the output of each position is a two-dimensional classification score representing a target and a background, can be represented by the following equation:(9)$$S=T_{z}*\;S_{Final}$$
where ∗ is a convolution operation uses the template features $T_{z}$ as the convolution kernel.

While obtaining the classification map S, the positions (*i*, *j*) in the map are mapped to the positions in the input search area by the following equation.
(10)$$[p_{i},p_{j}]=[\lfloor \frac{w_{im}}{2}\rfloor +(i-\left\lfloor \frac{w}{2}\right\rfloor )\times s,\lfloor \frac{h_{im}}{2}\rfloor +(j-\lfloor \frac{h}{2}\rfloor )\times s]$$
where $w_{im}$ and $h_{im}$ denote the width and height of the input search patch, and s denotes the total stride of the network. If the corresponding position $\left(p_{i},p_{j}\right)$ falls inside the ground truth bounding box, it is marked as a positive sample, and if it falls outside the box, it is marked as a negative sample. We choose cross-entropy loss as our classification loss $L_{\mathrm{cls}}$.

Immediately after, we analyze our regression branching network. Significant progress has been made in recent years in the feature representation of visual tracking, but the problem of feature inconsistency between classification and regression tasks has been largely ignored, and in most advanced trackers, feature-extraction methods have no effect on either task. Among our algorithm improvements, we believe that the features extracted from the salient regions provide more recognizable visual patterns for the classification task, and these features around the boundaries can contribute to the accurate estimation of the target state for the regression task.

We were inspired by CenterNet [46] in the field of object detection to model an object as a single point, the centroid of a bounding box, and then use estimation of the key points to find the centroid and regress to all other object properties, such as size, orientation, and even pose.

Our CenterNet network translates target feature learning into key point estimation. The target regression branch is the same as that in reference [47]. For details, please refer to the original text. Our classification branch loss and regression branch loss together constitute our base learner loss.
(11)$$L_{\mathrm{base}}\left(\theta \right)=\lambda \;L_{\mathrm{cls}}(\theta )+\left(1-\lambda \right)\;L_{\mathrm{reg}}(\theta )$$

Our base leaner achieves our goal by applying an iterative optimization algorithm. Here, we are mainly looking at our base learner as a linear model $b_{\theta }\left(x\right)=\theta x$, where θ is the classification weights. By doing so, faster convergence can be achieved. Our main approach is to use a strategy based on the Steepest Descent.

First a positive definite quadratic approximation is made to our function. Then a simple closed form expression is used to provide the optimal step size $\alpha^{\left(d\right)}$ in the gradient direction.
(12)$$\theta^{\left(d+1\right)}=\theta^{\left(d\right)}-\alpha^{\left(d\right)}\nabla \;L_{\mathrm{base}}(\theta^{\left(d\right)})$$
(13)$$\alpha^{\left(d\right)}=\frac{\nabla \;L_{\mathrm{base}}{\left(\theta^{\left(d\right)}\right)}^{T}\nabla \;L_{\mathrm{base}}\left(\theta^{\left(d\right)}\right)}{\nabla \;L_{\mathrm{base}}{\left(\theta^{\left(d\right)}\right)}^{T}H^{\left(d\right)}\nabla \;L_{\mathrm{base}}\left(\theta^{\left(d\right)}\right)}$$
where *d* is the iteration number, $H^{\left(d\right)}$ denotes the positive definite Hessian approxima-tion.

### 3.4. Offline Training

We train our model on the training splits of LaSOT and GOT-10k datasets. The backbone network was initialized with the weights from ImageNet. The proposed network structure is end-to-end trained in the offline training process. We trained 50 epochs by sampling 10,000 videos per epoch, for a total training time of less than 24 h on two Nvidia 2080Ti GPUs. (The training time here mainly refers to the total time from the beginning of training to the completion of model training under the condition of GPU acceleration and without the influence of other external factors.) Both the template branch and the search branch input are scaling the image to 512 × 512. Before getting the feature maps to input to the transformer, we added an additional convolutional layer (3 × 3 Conv + BN) to reduce the number of backbone feature channels from 1024 to 512. For the other regression branch, the image goes through resnet50 to extract features to get feature 1 size 1 × 2048 × 16 × 16, feature1 goes through the deconvolution module Deconv (i.e., as shown in Figure 1, the white block in front of CenterNet). The purpose of the deconvolution module is to input the characteristics of the image so as to output the image, which plays a role of restoration. By inputting the image into CenterNet, the state information of the target can be better regressed. Three times upsampling to get feature 2 size 1 × 64 × 128 × 128, send feature 2 into three branches for prediction, respectively, prediction heatmap size is 1 × 80 × 128 × 128 (indicating 80 categories), predicted length and width size is 1 × 2 × 128 × 128 (2 indicates length and width), predicted centroid offset size is 1 × 2 × 128 × 128 (2 indicates x, y). As an optimizer, we use SGD [47] with a momentum gradient descent Nesterov momentum of 0.9 and a weight decay of 0.0005. Learning rate set to 0.01. First of all, Nesterov momentum optimization algorithm is an improvement of momentum optimization algorithm. The difference is that the former applies the step of temporary update, that is, update the parameters with the current speed V first. After doing so, Nesterov algorithm momentum algorithm updates faster, because each update is based on the current speed and gradient. Then update the current speed before calculating the gradient, this will accelerate the convergence speed.

After training the network model, we can get the tracking results of video frames. The function of classification branch is to classify the location of the target from the images containing the target and background. Combined with regression branch, we can get the state information about the target, that is, the location of the center point and the boundary information, so as to get the final tracking results.

### 3.5. Online Tracking

We further equip our tracking algorithm with an online update model. Inspired by [23,24], we introduce an online branch to determine the change in the appearance of the target object during tracking. As shown in Figure 3 (bottom), the online branch inherits the structure and parameter network of the first three stages of the backbone network. The fourth stage maintains the same structure as the backbone network, but its initial parameters are obtained by pre-training. For the model update, we use the fast conjugate gradient algorithm [23] to train the online branch during inference. The weights of the prospect score map are: the online branch and the estimated prospect score map of the classification branch, and the IoUNet in [23,24] is not used in our model because we have separate regression branches. We refer the reader to [23,24] for more details.

## 4. Experimental Result

This section presents the results of our tracker on several tracking benchmark datasets and compares them with state-of-the-art algorithms. In this section, the algorithm of this paper is compared with other advanced algorithms. Experiments were conducted on OTB2015, VOT2018, LaSOT, and GOT-10k. The tracking algorithms in this paper were trained and evaluated using Python and Pytorch on a workstation with Ubuntu 18.04 and equipped with a GTX2080Ti graphics card. The results of the comparison experiments on each dataset are shown in Section 4.1, Section 4.2, Section 4.3 and Section 4.4. A comparative experimental analysis is performed so that the effectiveness of each component of our model can be evaluated. The running speed comparison experiments are performed in Section 4.5.

### 4.1. OTB2015

Wu et al. proposed the Object Tracking Benchmark OTB to evaluate the performance of single visual-object tracking algorithms based on the previous work. The OTB not only provides evaluation metrics for evaluating visual-object tracking algorithms, but also provides some labeled challenging video sequences, which have been expanded from the original 50 to the current 100 video sequences and include various common challenges. The OTB benchmark also provides both evaluation toolkits, including MATLAB and Python versions, and has a simple function interface with good ease of use, making the OTB widely used.

The OTB evaluates the performance of the tracking algorithm by two main evaluation metrics: precision rate based on the center position error and success rate based on the overlap rate.

Our proposed algorithm is evaluated in this OTB2015 dataset to check its performance. Ten representative tracers were selected for the experiments: SiamFC [26], SiamRPN [29], SiamRCNN [48], SiamCAR [33], SiamAttn [49], SiamBAN [31], Ocean [32], DiMP50 [24], PrDiMP50 [25], TransT [34], and our baseline TrDimp [7]. The above trackers are all from recent popular tracking algorithms and are used for comparison with our proposed algorithm.

In the OTB2015 test, the performance of the tracking algorithm was evaluated using one-pass evaluation (OPE). The overlap rate (OR) and central localization error (CLE) were applied to obtain the respective success rate and accuracy plots. For the success rate plot, the value in square brackets indicates the area under the curve (AUC) value; for the accuracy plot, the value in square brackets indicates the score value for an error threshold of 20 pixels. Figure 4 shows the precision and success plots of the experimental results for all 100 videos.

As the results in Figure 4 show, it appears that our algorithm achieves the best performance based on the comparison of the AUC scores. This indicates that our algorithm can further improve the robustness of the baseline with the resulting overall AUC score of 0.715 and an overall AUC score improvement of 0.9% compared to TrDimp.

In OTB benchmark, all video sequences involve a total of 11 different difficult attributes faced by tracking, including: illumination variation (IV), scale variation (SV), occlusion (OCC), deformation (DEF), motion blur (MB), fast motion (FM), in-plane rotation (IPR), out of plane rotation (OPR), out of view (OV), background clutters (BC), and low resolution (LR). For each video sequence, it contains many attributes mentioned above. To evaluate the performance of the algorithm in this chapter under different attributes, experiments were conducted on OTB-2015 in this chapter, and the success rate plots for different attributes are shown in Figure 5, using the success rate of the challenge attributes to measure the performance of the tracking algorithm in handling specific challenges. It can be seen from the figure that our proposed algorithm can handle challenges for almost all attributes, although it does not outperform other algorithms on every attribute, indicating that our proposed algorithm has good robustness. As shown in Figure 5, the accuracy plots and the success plots for different challenge attributes are demonstrated.

As can be seen from Figure 5, the tracker we proposed does not perform well in the three attributes of OV, MB, and LR. For these three attributes, out of view (OV) mainly refers to some portion of the target leaves the view, motion blur (MB) meant the target region is blurred due to the motion of target or camera, and low resolution (LR) represents the number of pixels inside the ground-truth bounding box is less than 400. In contrast, the reason why our tracker does not perform well on several of the above metrics seems to be because the bounding box of the object is not accurately regressed. The good performance of our tracker in the remaining properties shows the effectiveness of our proposed network structure.

### 4.2. VOT2018

VOT is a target tracking competition that has been held annually since 2013 and generally serves as a workshop for the ICCV and ECCV conferences. Starting from VOT2016, three main metrics are used to measure the performance of target tracking algorithms.

(1) Accuracy: The overlap between the predicted target bounding box and the manually marked target bounding box is calculated for the tracking box predicted by the target tracking algorithm in the test video, and the performance of the algorithm is measured by the degree of overlap of the bounding box; the higher the overlap rate, the better the accuracy of the target tracking algorithm;

(2) Robustness: The target tracking algorithm may not succeed in a single run after the test video, and it may need several re-initializations before it succeeds, and the target tracking algorithm loses the tracking target during the running process, then it needs to reinitialize to ensure the continuous tracking of the target. The number of times the tracking algorithm needs to re-initialize the tracking algorithm to complete the tracking task of a complete video sequence is counted, and the lower the number, the better the robustness of the algorithm;

(3) Expected Average Overlap Expectation (EAO): In short-term image sequences, the appearance of the target will change due to occlusion and other conditions, resulting in the loss of the target in the tracking process. The robustness of the algorithm is measured by the number of initialization times of resetting the algorithm. However, EAO does not restart the algorithm after the target tracking fails, the tracking accuracy of the algorithm is characterized by calculating the expected overlap between the target bounding box predicted by the target tracking algorithm and the actual bounding box. The larger the value, the higher the accuracy of the target tracking algorithm. The results of the comparison experiments are shown in Table 1.

Our proposed algorithm is compared with the nine latest trackers on VOT2018, as shown in the results in Table 1. Our algorithm achieves a leading performance compared to these latest trackers. SiamFC [26], SiamRPN [29], SiamRCNN [47], SiamAttn [48], SiamBAN [31], Ocean [32], DiMP50 [24], PrDiMP50 [25], and our baseline TrDimp [7] are listed in Table 1. As shown in the data in the form, we marked the first and second place of the tracker with good performance in red and blue respectively. Our tracker gets an EAO score of 0.489, which is far better than the advanced trackers SiamRCNN, SiamAttn, and other latest trackers. Its performance is equal to that of the state-of-the-art Ocean. It outperforms SiamAttn by 2.9% and TrDiMP by 3.6%. Achieving huge gains in target tracking on challenging benchmarks. Although it is not optimal in terms of accuracy and robustness, it still has great improvement compared with our baseline algorithm.

### 4.3. LaSOT

The LaSOT dataset is a joint effort between Temple University in the U.S.A., South China University of Technology in China, and Pengcheng Lab. LaSOT consists of 1400 sequences with over 3.5 million frames in total. Within these sequences, each frame is carefully and manually bounding box annotated, the average video length of LaSOT is over 2500 frames, and each sequence contains a variety of challenges from the field, where targets are characterized by the possibility of disappearing and reappearing in the view.

As much as possible, LaSOT dataset provides the same number of videos for each category of targets, which prevents category imbalance during training.

To enable further analysis of tracker performance, the LaSOT dataset is tagged with 14 attributes for each sequence, including Illumination Variation (IV), Full Occlusion (FOC), Partial Occlusion (POC), Distortion (DEF), Motion Blur (MB), Fast Motion (FM), Scale Variation (SV), Camera Motion (CM), Rotation (ROT), Background Clutter (BC) Low Resolution (LR), Viewpoint Change (VC), and Line of Sight (OV). The most common challenging factors in LaSOT are scale change (SV and ARC), occlusion (POC and FOC), distortion (DEF), and rotation (ROT), respectively, which are well-known challenges in tracking real-world applications.

As in OTB2015, LaSOT performs a once-through evaluation (OPE) and measures the accuracy of different tracking algorithms under three metrics. Accuracy is calculated by comparing the distance (in pixels) between the tracking result and the true value bounding box. Using a normalized accuracy approach, the LaSOT dataset ranks the trackers using scores with AUC between 0.5. The success rate is calculated as the overlap between the rectangular bounding box obtained from the tracking prediction and the true value bounding box. The tracking algorithm uses an AUC between 0 and 1 for ranking.

Figure 6 shows a plot of the tracking results on the LaSOT benchmark in comparison with the results from the latest tracker TransT [34], SiamRCNN [47], SiamCAR [33], SiamBAN [31], TrDimp [7], and SiamDW [44]. As the graph shows, our tracker also performs well. In order to further evaluate our proposed model, as shown in Figure 6, showing the results on LaSOT, our tracker achieved a SUC score of 0.659, which exceeded 0.02 and 0.01 compared with TrDiMP and TransT, respectively. Among all the comparison methods, our tracker achieves the best normalized precision score of 0.738.

### 4.4. GOT-10k

The GOT-10K dataset is composed of 10,000 videos in WordNet. Its purpose is to provide a unified training and evaluation platform for developing trackers with rich motion trajectories. These sequences are classified into a total of 563 classes of moving objects, six tracking attributes and 87 classes of motion to cover as many challenging patterns as possible in real-world scenes.

GOT-10K is divided into three parts: training, validation, and testing. The training part contains 9340 sequences with 480 object classes, while the test part contains 420 videos with 83 object classes and an average length of 127 frames per sequence.

The tracking results on the GOT-10k benchmark compared to the results from the latest tracker are presented in Table 2. As the table shows, our tracker also performs well. As shown in the results in Table 2, the trackers with the best performance and the second best performance on different indicators are marked with red and blue, respectively. The proposed tracker model achieves the AO score of 0.667. Although our proposed algorithm has a lower AO score compared to the baseline algorithm, our tracker still has better performance compared to other excellent algorithms.

### 4.5. Running Speed Comparison Experiment

While the experimental sections above are all evaluating the robustness of the tracker, the tracking speed is another very important metric to evaluate the tracker, especially to meet the real-time requirements. However, evaluating the tracking speed is not straightforward because there are many key factors influencing it, as explained in the introduction section. To reduce the influence of hardware, the VOT2014 committee introduced a new unit, the Equivalent Filter Operation (EFO), which is based on a predefined filtering operation that is automatically performed by the toolkit before running the experiment to report the tracking speed. Figure 7 illustrates the tracking speed of the SOTA tracker in terms of EFO on a VOT2018 basis.

The comparison graph of the results shows that our proposed algorithm can guarantee both accuracy and real-time performance, which not only improves the robustness of the tracker, but also achieves faster tracking efficiency.

As for whether the method proposed in this paper reduces the number of times required for training, it can be explained from two aspects. First, the cost of training time. Compared with the baseline algorithm TrDimp, the original author pointed out that he used four 1080Ti to train 50 epochs for a total of more than 30 h. We used two 2080Ti to train 50 epochs, and only used nearly 24 h; second, real-time performance. The tracking rate of our proposed algorithm can reach close to 60fps, which shows that the model we train can meet the real-time performance while guaranteeing the tracking accuracy.

## 5. Conclusions

In order to improve the robustness and real-time performance of single-target visual object tracking, this paper proposes a twin network based on a meta-optimization algorithm, in which instead of a single target, the template branches are several initial frames integrated into a concatenated network, while a transformer is used to learn the appearance representation of the target. The entire network utilizes the idea of meta-learning, including a meta-learner module, and a base-learner module, using a meta-optimization approach which allows for fast iteration. In addition, a template update strategy is designed to ensure the effectiveness of the templates in the inference phase. Performance comparisons on several challenging benchmarks demonstrate the effectiveness of our proposed approach. It not only leads to significant performance improvements, but also helps the tracker to be used in real-world tracking scenarios. However, there are some issues, such as long-term tracking performance degradation, which still need to be addressed.

In conclusion, the proposed method has room for development and improvement. Future research should focus on exploring meta-learning and video image segmentation strategies for object visual tracking.

## Figures and Tables

**Figure 1 sensors-22-05826-f001:**
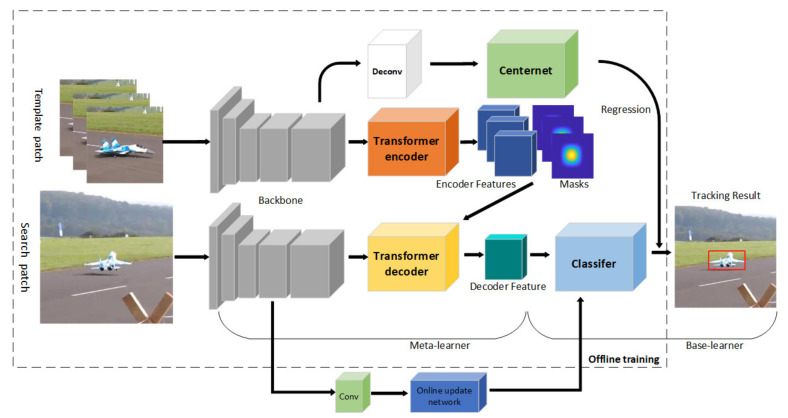
The overall architecture of our proposed single object visual tracking algorithm.

**Figure 2 sensors-22-05826-f002:**
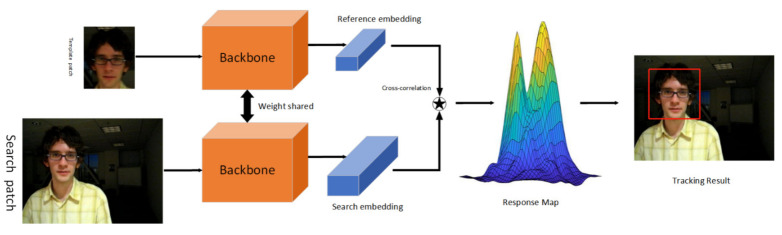
The Siamese tracking pipeline for generic object tracking.

**Figure 4 sensors-22-05826-f004:**
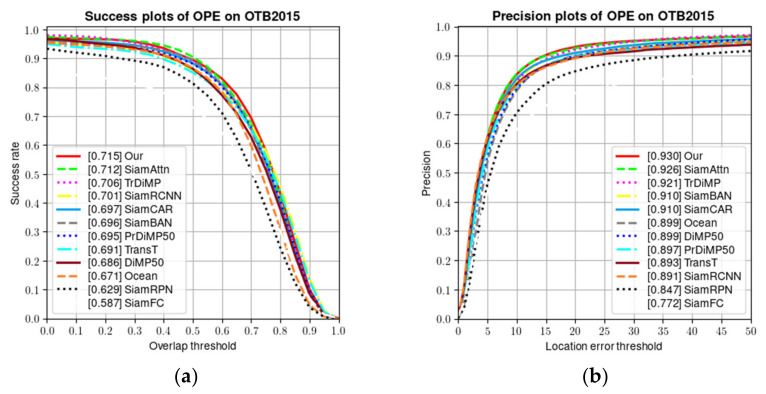
Comparison of the success rate and accuracy of the proposed algorithm with baseline algorithm and other advanced tracking algorithms. (**a**) success plots; and (**b**) precision plots.

**Figure 5 sensors-22-05826-f005:**
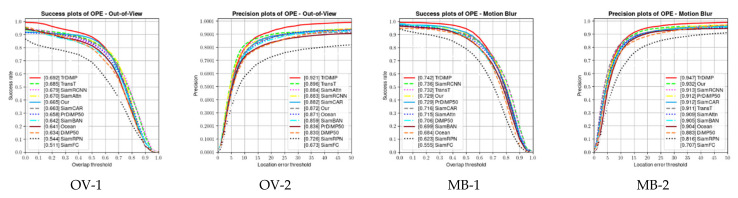
Precision plots and success plots corresponding to some typical scenarios. The OTB2015 dataset was used for the evaluation.

**Figure 6 sensors-22-05826-f006:**
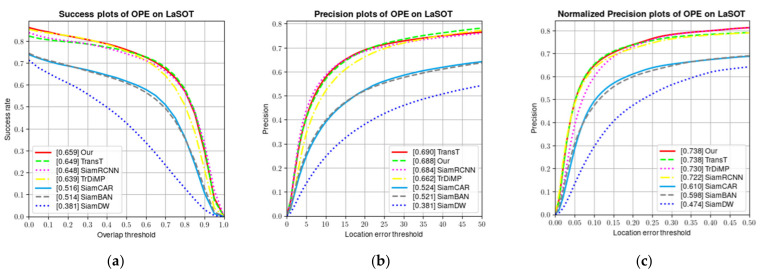
Comparison of the success rate and accuracy of the proposed algorithm with baseline algorithm and other advanced tracking algorithms. (**a**) Success plots; (**b**) precision plots; and (**c**) normalized precision plots.

**Figure 7 sensors-22-05826-f007:**
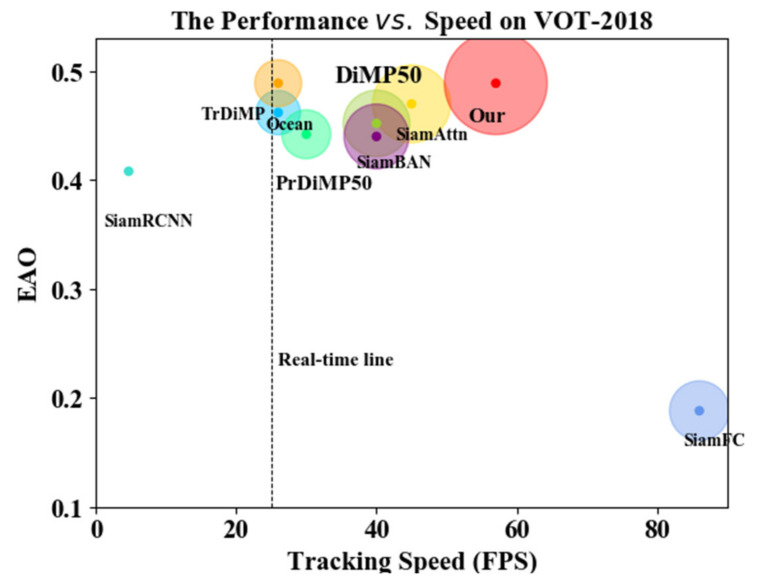
Comparison of the results of the proposed algorithm with other algorithms in terms of running speed and EAO.

**Table 1 sensors-22-05826-t001:** EAO score ranking of trackers tested at VOT-2018.

Tracker	Accuracy	Robustness	EAO
SiamFC	0.503	0.585	0.188
SiamRPN	0.490	0.464	0.244
SiamR-CNN	0.609	0.220	0.408
SiamAttn	0.630	0.160	0.470
SiamBAN	0.597	0.178	0.452
Ocean	0.592	0.117	0.489
DiMP50	0.597	0.153	0.440
PrDiMP50	0.618	0.165	0.442
TrDiMP	0.600	0.141	0.462
Our	0.625	0.119	0.489

**Table 2 sensors-22-05826-t002:** AO score ranking of trackers tested at GOT-10k.

Tracker	AO	$SR_{0.5}$	$SR_{0.75}$
SiamFC	0.392	0.426	0.135
SiamRPN	0.481	0.581	0.270
SiamR-CNN	0.649	0.728	0.597
Ocean	0.611	0.721	-
DiMP50	0.611	0.717	0.492
PrDiMP50	0.634	0.738	0.543
TrDiMP	0.671	0.777	0.583
Our	0.667	0.725	0.599

## Data Availability

The data used to support the findings of this study are available from the corresponding author upon request.

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
