# Peer review of "Fast and Robust Visual Tracking with Few-Iteration Meta-Learning"

_sensors, 2022, doi:10.3390/s22155826_

Round 1

Reviewer 1 Report

This article is very well written with sections 1 and 2 offering valuable information to the reader. The figures are well integrated into the text and the references are relevant and up-to-date. The proposed tracking method presented by the authors appears very useful and holds significant improvements in real-world tracking scenarios. It also leaves room for future improvements which even though mentioned are not examined. Some changes in the headings i.e experiments->results.

Reviewer 2 Report

This paper presents a neural network based visual object tracking system. The algorithm uses a Siamese Network-based approach, and it has some enhancement using transformer architecture. It also has an online update model. The system achieves a good balance between accuracy and speed.

This paper conducts extensive evaluation, but it does not summarize the results clearly. For example, regarding the accuracy of the algorithm on challenging scenarios, how is its performance compared to others? I would suggest adding a short summary in the abstract, and a detailed explanation of the results in the evaluation section.

There are some writing issues as the following:

Section 2.1

The last paragraph is confusing.

Section 2.3

"design a new online update method that improves faster FPS" -> achieves higher FPS?

Section 3.3

"Our base leaner is achieves our goal" -> "Our base learner achieves our goal"

Section 4.1

The second paragraph of this section is repetitive.

"Our proposed algorithm is further evaluated". We can remove "further" here.

"In the OTB benchmark test, all sequences are labeled with 11 difficult attributes, each of which contains several attributes." This sentence is confusing.

Both OTB2015 and OTB-100 are used in this section. Are they the same thing?

There are many subfigures in Figure 5, but this section lacks a clear summary of the results.

Section 4.2

The definition of "Expected Average Overlap Expectation (EAO)" is confusing. Could you revise it?

Section 4.3

Figure 6 needs adjustment to make sure the caption and the figure are on the same page.

Section 4.4

"The graphs of tracking results on the GOT-10k benchmark compared to the results from the latest tracker are presented in Table II." What are graphs? And I propose to use "Table 2" to be consistent with other parts of the paper.

"The algorithms for the comparison are the following: SiamFC [26], SiamRPN [29], SiamRCNN [46], Ocean [32], DiMP50 [24], PrDiMP50 [25], TrDimp [7]." This sentence can be removed.

How are the results of the proposed algorithm compared to others?

Section 4.5

I assume the "Figure 8" on section 4.5 means "Figure 7" because I am not able to find Figure 8 in this paper. What does the area of the circles mean in this figure?

Section 5

"It not only leads to significant performance improvements. It also helps" -> ", but also helps".

Reviewer 3 Report

This paper mainly adds a meta-learning mechanism to a visual tracking algorithm described in a CVPR 2021 paper (ref.[7]).
In general, in my opinion the authors shuld expand the description of their new modifications.
An extensive experimental section report and compare their results.

Major comments

Pag.3 last paragraph of 2.1 section. You write that SN trackers outperform DCF, but this is not really true (for instance [24]).
The Figure 2 of the overall architecture is neither referred to nor commented on. For instance, what is the white block just before the Centernet block? Please explain how the tracking result is obtained. Where is the cross-correlation operation of Figure1?
Figure 3 in Pag.6 should be re-drawn, to write the inputs of self-attention and to fix the M input  or simply referred to that shown in [7]. If re-drawn, you should also describe the Key, Query, Value.
The implementation details in section 4 should be extended. For example, how the training is performed? what is the training time? what is the NN optimization algorithm?
as regards the experimental results, if the meta-learning aims to reduce the number of instances needed for training, how this is experimentally shown?
Furthermore, I find confusing the pages full of graphs. You could separate the couple of graphs for each attribute maybe adding some comment for each attribute, which extends the number of pages but it makes the results clearer. Another possibility is reduce the number of graphs by using some more compact measure such as for instance the F-score or the F-measure.

Minor comments

Pag.2, 1st paragraph. I suggest to delete the conjunction 'as' after the world 'Recently' so that the sentence makes sense. Moreover, I suggest to delete the last phrase of the paragraph (We have therefore lanched our research...) because in my opinion it is not appropriate.
At the end of the next paragraph, please add the full description of the acronym SOTA before the acronym, and put SOTA within round brackets.
Please add the description of the paper structure at the end of the Introduction.

Pag.3, last paragraph of 2.1. Please complete the list of limitations of SN-based trackers, not only '...and so on...'.
Pag.3, first paragraph of 2.2 section, after '... is known as meta-learning....' please add a reference, such as A perspective view and survey of meta-learning, by R. Vilalta & Y. Drissi.

Pag.6, 2nd paragraph, add 'are' between 'inputs' and 'pairs'.
In my opinion, the next paragraph should be rearranged to give a clearer description. For example:
"In template branch, several frames (i.e., the template image patch ?∈ℝ^{3×?0×?0}) are first passed to the backbone network to obtain their features maps ??∈ℝ^{?×?×?}.
The features map are then fed to the encoder for feature representation."
The next phrase should be: "The basic block of the encoder is the self-attention mechanism."
At this point, please add a reference to the self-attention mechanism, with a short description such as: "such mechanism aims to mutually enforce the features from multiple templates".
In the next sentence, please add 'A_{T->T}' after '...similarity matrix' and 'matrix' after '...the A_{T->T}'.
Pag.7, just after eq.(6), 'featrure' -> 'feature'.
Pag.8, after (11), delete 'is' after 'learner' and delete 'that' after 'goal'.

Round 2

Reviewer 2 Report

Thanks for the authors' efforts to revise the draft. I reviewed the revised sections and they are improved. I still found some issues though. Probably the authors need to proofread the whole paper again after fixing them to make sure the issues are properly addressed.

Some writing issues are the following:

Section 2.1

"but the tracker based on siamese network has attracted our attention because it can balance accuracy and real-time." -> balance accuracy and speed.

Section 3.1

"After a pair of pictures pass through two branches to get a feature map" -> seems better to replace "pass" with "go".

Section 4.1

"After analysis, we think the reason why these attributes don't perform well should be the problem of the estimation of the target state.". This sentence seems strange.

Section 4.2

"As shown in the data in the form, we marked the first and second place of the tracker with good performance in red and blue respectively." We had better merge this paragraph with the previous one.

Section 4.3

"In order to further evaluate our proposed model, as shown in Figure 6…." We had better merge this paragraph with the previous one.

Section 4.4

"As shown in the results in Table 2, the trackers with the best performance and the second best performance…" We had better merge this paragraph with the previous one.

"Although compared with baseline algorithm, it reduces 0.004 points, compared with other algorithms, it still has certain competitiveness." Please revise this sentence.

Section 4.5

"Our proposed algorithm can achieve a frame rate close to 60fps on the premise of ensuring tracking accuracy, which shows that our trained model can converge quickly." I guess you mean running the inference faster?

Reviewer 3 Report

The important points seem to be fixed.

I recommend the authors accurately read the entire manuscript to fix possible typos.

Author Response

Dear Editors and Reviewers:

We would like to express our heartfelt thanks for your affirmation of the revision of the article. Based on your second comments and the other reviewers, we have made the final revisions to the article, marked in red, and finally, again, express our thanks.